# Construction and Characterization of a High-Capacity Replication-Competent Murine Cytomegalovirus Vector for Gene Delivery

**DOI:** 10.3390/vaccines12070791

**Published:** 2024-07-18

**Authors:** André Riedl, Denisa Bojková, Jiang Tan, Ábris Jeney, Pia-Katharina Larsen, Csaba Jeney, Florian Full, Ulrich Kalinke, Zsolt Ruzsics

**Affiliations:** 1Medical Center, Institute of Virology, University of Freiburg, 79104 Freiburg, Germanyflorian.full@uniklinik-freiburg.de (F.F.); 2Faculty of Medicine, University of Freiburg, 79104 Freiburg, Germany; 3Institute of Medical Virology, Goethe University Frankfurt, University Hospital, 60596 Frankfurt am Main, Germany; 4TWINCORE, Centre for Experimental and Clinical Infection Research, a Joint Venture between the Hanover Medical School and the Helmholtz Centre for Infection Research, Institute for Experimental Infection Research, 30625 Hanover, Germany; 5Department of Microsystems Engineering—IMTEK, University of Freiburg, 79110 Freiburg, Germany

**Keywords:** cytomegalovirus, host range, cross-species application, gene transfer vector, genetic stability, high-capacity vector

## Abstract

We investigated the basic characteristics of a new murine cytomegalovirus (MCMV) vector platform. Using BAC technology, we engineered replication-competent recombinant MCMVs with deletions of up to 26% of the wild-type genome. To this end, we targeted five gene blocks (m01-m17, m106-m109, m129-m141, m144-m158, and m159-m170). BACs featuring deletions from 18% to 26% of the wild-type genome exhibited delayed virus reconstitution, while smaller deletions (up to 16%) demonstrated reconstitution kinetics similar to those of the wild type. Utilizing an innovative methodology, we introduced large genomic DNA segments, up to 35 kbp, along with reporter genes into a newly designed vector with a potential cloning capacity of 46 kbp (Q4). Surprisingly, the insertion of diverse foreign DNAs alleviated the delayed plaque formation phenotype of Q4, and these large inserts remained stable through serial in vitro passages. With reporter-gene-expressing recombinant MCMVs, we successfully transduced not only mouse cell lines but also non-rodent mammalian cells, including those of human, monkey, bovine, and bat origin. Remarkably, even non-mammalian cell lines derived from chickens exhibited successful transduction.

## 1. Introduction

Several platforms for recombinant vaccines offer potent neutralizing antibody responses against prevalent targets. However, some challenges and important targets, such as the clearance of persistent pathogens, the prevention of escape through high epitope variability of surface proteins, or the induction of an anti-tumor response, remain to be met since they additionally require strong cell-mediated immune responses [1,2]. Currently, recombinant cytomegaloviruses (CMVs) are intensively explored as vaccine vectors [3,4,5,6,7]. CMVs excel at provoking the strongest clinically known T cell responses in their natural hosts and, consequently, came into focus as a viral vector platform with a special focus on these targets [8]. CMV infection induces specific CD8^+^ T cell responses, which can account for 5% of the entire CD8^+^ T cell population and eventually result in an effector memory phenotype [9,10]. While these cells circulate and reside in peripheral tissues upon antigen encounter, they can respond ad hoc [11,12,13,14]. In addition to cellular immunity, the humoral response elicited against murine cytomegalovirus (MCMV) in mice exhibits an extended duration after infection [15]. Interestingly, even attenuated variants of MCMVs and rhesus CMVs (RhCMVs) still evoke this characteristic immune response in their respective hosts [16,17]. Further, the immunomodulatory properties of CMVs enable superinfection, which allows not only strategies relying on a homologous prime/boost vaccination regime but simultaneously reduces anti-vector immunity hazards [18,19,20]. Experimental recombinant human CMV (HCMV) vaccines aim at the prevention of infections with the wild-type virus, while recombinant RhCMVs and MCMVs are used experimentally to target orthologous pathogens or as tumor vaccines. It is believed that CMVs’ broad in vivo tropism, including antigen-presenting cells and inflammatory macrophages, facilitates therapeutic applications [21]. The genomes of CMVs rank among the largest of all known mammalian viruses, and they comprise double-stranded DNA genomes larger than 230,000 base pairs in length [22]. Combined with the fact that many encoded genes are dispensable for virus production in vitro, CMV vectors may provide a substantial theoretical payload capacity, enabling the delivery of either a large singular transgene or multiple smaller transgenes with the potential for simultaneous or coordinated expression [23,24]. In addition, the genome structure of MCMVs conveys the convenience of using their potential for extraordinary vector capacity. With short terminal repeats of only 31 bp, they diverge from other herpesviruses, which carry large terminal and internal repeats in the range of many kilobase pairs (kbp) in length. The central genome region of MCMVs is populated by core genes that are essential for lytic virus replication—in vitro and in vivo—and, therefore, are conserved among most herpesviruses. Moving from the center towards the terminal sites, genes conserved within the CMV family or β-herpesviruses are located. Finally, large host-specific accessory gene clusters can be identified in the most terminal regions; they are primarily responsible for the suppression and modulation of host immunity and, therefore, are dispensable for virus propagation in vitro [25,26,27,28] (see Figure 1a). MCMVs encode 170 annotated genes, of which 101 are non-essential for viral growth in vitro and are accumulated in gene blocks toward the genome ends. This cumulative non-essential gene pool contributes to an extraordinary theoretical payload capacity of nearly unmatched proportions, totaling 70 kbp. An established in vitro cell culture system in which recombinant MCMV vectors can be proficiently propagated to high-titer virus stocks renders this platform feasible for therapeutic use [27,29].

Currently, two key questions need to be clarified to use MCMVs as vectors for immunization in relevant medical or veterinary hosts: (i) whether recombinant MCMVs can efficiently deliver transgenes to cells of species other than mice and (ii) how much of their potential for transgenes can be used while maintaining stable vector propagation. Nevertheless, in the use of other viral vector platforms in a cross-species context, administration resulted in a successful therapy, and representative vectors from poxviridae and Adenoviridae were named ALVAC and ChAdOx-1, respectively [30,31]. Particularly for CMVs, an RhCMV-based vector proved its vaccination potential in a cross-species context, ultimately resulting in therapeutic success [32]. Moreover, there is evidence that RhCMVs—in vitro—are capable of infecting cells from different species, including humans [33]. Additionally, indications suggest that recombinant MCMVs exhibit potential applications in species beyond mice. Observations include successful viral entry and viral gene expression in rat fibroblasts, hamster cells, and even cells of human and monkey origin [34,35,36,37]. 

In this investigation, we evaluate the cross-species applicability of MCMV vectors, showcasing their successful high-efficiency transgene expression in an unexpectedly and unprecedentedly broad host range. We also assess the vectors’ replication capacity in non-host cells. Our study introduced an MCMV-based platform with a practical payload capacity surpassing 46 kbp. Notably, the platform accommodates and sustains a minimum of 35 kbp of foreign DNA without discernible impairment in viral fitness in vitro or genetic instability. Additionally, we conducted a comparative transcriptome analysis of host and non-host cells, examining the wild-type MCMV and recombinant vector transcriptomes in a kinetic manner.

## 2. Materials and Methods

Cell lines and *E. coli* strains. ARPE-19 (ATCC CRL-2302), A549 (ATCC CCL-185), SeBu [38] (HFF), 293A (a sub-clone of HEK293 [39], ThermoFisher, Waltham, MA, USA), HeLa (ATCC CCL-2), 911 [40], Vero E6 (ATCC CCL-81), CV-1 (ATCC CCL-70), NIH 3T3 (ATCC CRL-1658), PK-15 (ATCC CL-101), VR1BL [41], EpoNi [42], MDBK (ATCC CCL-22), DF-1 [43], and LMH [44] cells were cultured in Dulbecco’s modified Eagle’s medium (DMEM; Gibco, New York, NY, USA) supplemented with 10% (*v*/*v*) fetal bovine serum (FCS; Merck, Darmstadt, Germany) and 100 U/mL penicillin/streptomycin (Pen/Strep; Gibco, USA). MDCK (ECACC 84121903) cells were cultured in DMEM supplemented with 5% (*v*/*v*) FCS and Pen/Strep. Mouse embryonic fibroblasts (MEFs) were prepared from 14–15-day-old fetuses of Balb/c mice, excluding the brains, eyes, and liver. The fibroblasts were cultured in DMEM with 10% (*v*/*v*) FCS and 100 U/mL Pen/Strep. 

We used the *E. coli* Neb10beta strain (genotype: Δ(ara-leu) 7697 araD139 fhuA ΔlacX74 galK16 galE15 e14-ϕ80dlacZΔM15 recA1 relA1 endA1 nupG rpsL (StrR) rph spoT1 Δ(mrr-hsdRMS-mcrBC)) for BAC maintenance and mutagenesis, the *E. coli* Neb5alpha strain (genotype: fhuA2 Δ(argF-lacZ) U169 phoA glnV44 Φ80 Δ(lacZ)M15 gyrA96 recA1 relA1 endA1 thi-1 hsdR17) for plasmid amplification (both from New England Biolabs (NEB), Ipswich, MA, USA), and the *E. coli* Pir-1 strain (genotype: F-∆lac169 rpoS(Am) robA1 creC510 hsdR514 endA recA1 uidA(∆MluI)::pir-116) for the amplification of plasmids with R6Kγ as the origin of replication (ThermoFisher, USA).

Viruses. All MCMVs used in this study were BAC-derived and reconstituted in MEF by transfecting 2.5 µg of purified BAC DNA into approximately 1 × 10^5^ MEF cells using the Superfect transfection reagent (Qiagen, Hilden, Germany) according to the manufacturer’s instructions. Cells were monitored for plaque formation daily. The plaques occurred 3–10 days post-transfection. All viruses were passaged on MEF six times, and virus stocks were purified on a sucrose cushion as described earlier [45]. Wild-type MCMV (MCMV-wt) was derived from the bacterial artificial chromosome (BAC) of pSM3fr-MCK-2fl, as described elsewhere [46]. For details on the construction of the MVMV-BACs, the reader should refer to the Appendix A.

Molecular cloning. The detailed workflow for all plasmids and BACs constructed in this study is described in the Appendix A. The sequences of all oligonucleotide primers can be found in Appendix A. In addition, we generated overview tables listing the basic features of all the plasmids and BACs that we used or constructed in this study; these are presented in Appendix A, respectively. In this study, the recombinant MCMV-BACs with single deletions were derived from pSM3fr [47], and those with multiple deletions were derived from the parental BACs ΔI+II-MCMV [27] through further deletions using λ-red recombination (recombineering) while employing pKD46 as an inducible expression vector for the λ-red recombinases, as described earlier [48]. Red recombination was followed by marker removal by applying site-specific recombinases, as described earlier [49]. For the insertion of complex genetic features, we used 3SR, as described previously [49].

Comparative growth kinetics. MEF, HFF, ARPE-19, A549, 293A, 911, CV-1, Vero E6, MDBK, and DF-1 (5.0 × 10^5^ cells/well) in a 24-well format were infected with MCMV variants (MCMV-wt, ΔI+II-GLuc, Q4, Q4-LAD, and Q4-LRBAs-GLuc) at a low MOI (0.1), and the inoculums were incubated for 1 h on the cells. Directly after infection, the cells were washed three times with 1 × PBS (phosphate buffered saline; Gibco, USA) and cultured in DMEM with 10% (*v*/*v*) FCS and 100 U/mL Pen/Strep. Supernatants were harvested on each designated day up to day 7, clarified via centrifugation, and stored at −80 °C. The inoculum (day 0) and all collected supernatants were later titrated in parallel using a plaque assay, as described earlier [50]. The assays were carried out in technical triplicates and repeated three times.

Transgene expression. MEF, HFF, ARPE-19, A549, 293A, 911, HeLa, CV-1, Vero E6, MDBK, VR-1BL, PK15, MDCK, EpoNi, LMH, and DF-1 cells were seeded at 16,500 cells/well in a 24-well format. Four hours post-seeding, cells were infected with MCMV variants (MCMV-wt, ΔI+II-GLuc, Q4-LRBAs-GLuc) at an MOI of 3. At 1 hpi, cells were washed and cultured in DMEM with 10% (*v*/*v*) FCS and 100 U/mL Pen/Strep. Supernatants were harvested at 24 hpi and at 2 days and 5 days post-infection, centrifuged, and quantified for their Gaussian luciferase activity. GLuc activity was measured at a designated time point described earlier [51] to determine the background at 1 hpi. The assays were carried out in technical triplicates and repeated three times. 

Viral genome quantification using quantitative PCR (qPCR). MEF, ARPE-19, A549, 293A, 911, and Vero E6 cells (2.5 × 10^5^ cells/well) in a 24-well format were infected at an MOI of 3 with either ΔI+II-GLuc or Q4-LRBAs-GLuc. Cells were harvested at 14 hpi, 31 hpi, 3 dpi, and 5 dpi. DNA was isolated using the NucleoSpin Tissue kit (Macherey-Nagel, Dueren, Germany), and 10 ng of DNA was used for duplicated qPCR reactions with the QuantiTect SYBR Green PCR kit (Qiagen, Hilden, Germany). Viral DNA counts were quantified by calculating the ΔΔCT threshold to compare the viral and host genomes. The M45 gene of the MCMV was amplified using M45for (5′-ATCTCCTCGAAGGGGAATGA-3′) and M45rev (5′-TCGACAGACAGCCGTTCGT-3′) [52]. To assess the viral genome/host ratio, GAPDH (glyceraldehyde-3-phosphate dehydrogenase) amplicons were generated using mugGAPDHfor (5′-CTGCAGTACTGTGGGGAGGT-3′) and mugGAPDHrev (5′-CAAAGGCGGAGTTACCAGAG-3′) [53] for MEF. For the amplification of human GAPDH, the primers hugGAPDHfor (5′-GGACTGAGGCTCCCACCTTT-3′) and hugGAPDHrev (5′-GCATGGACTGTGGTCTGCAA-3′) were used [54], and for Vero E6 cells, agmgGAPDHfor (5′-GTCTTTGCTGTCGTATGGGGG-3′) and agmgGAPDHrev (5′-CCTGGGGACTAGGGAAGGAAG-3′) were used.

Next-generation sequencing. The MEF cells (70% confluence) were infected with MCMV-wt, Q4, Q4-LAD, and Q4-LRBAs-GLuc at an MOI of 0.5. After 48 hpi, viral particles were harvested and purified, and DNA was extracted using the NucleoSpin Tissue kit (Macherey-Nagel, Germany). Illumina NGS was performed on a 100 ng DNA template for each sample (paired-end sequencing, 2 × 150 bp, 5 million reads/sample) at Eurofins (Germany). We employed the eautils and seqtk tools to preprocess the paired-end reads, which included merging, quality trimming, and conversion into FASTA format for analysis. To assess the coverage of the MCMV genome, strict GenBank (gb) files served as references. A custom Python script segmented the sequence into 50-base-pair (bp) fragments, which were formatted for the blastn tool. The parameters included the following: -evalue 10, -word_size 30, -num_threads 4, and -outfmt “6 qseqid sseqid pident qlen length mismatch gapope evalue bitscore. The resulting blast output file determined the coverage by aggregating based on sseqid. For further analysis, the MATLAB Bioinformatics Toolbox was applied, and the previously proposed Bayesian four-state Hidden Markov Model (HMM) [55] was employed. The four states in this model encompassed copy number loss, copy number neutrality, single copy number gain, and multiple copy number gain. Informative priors were integrated into the model, facilitating Bayesian learning from the available data. Posterior inferences were subsequently drawn to identify copy number gains and losses.

Transcriptome kinetic (RNA-seq) analysis. MEF, ARPE-19, A549, and 911 cells (5.0 × 10^5^ cells/well) in a 24-well format were infected 4 h post-seeding with MCMV-wt or Q4-LRBAs-GLuc at an MOI of 3. Harvesting occurred at 8 hpi and 31 hpi through centrifugation at 1000× *g* for 5 min. Cell pellets were washed with PBS, re-suspended in 350 µL of RLT buffer, and processed using the RNeasy Mini kit as per the manufacturer’s instructions in independent triplicates. At least 1500 ng of RNA in a 30 µL volume was isolated and sent for Illumina next-generation sequencing, resulting in paired-end sequencing with a read length of 2 × 100 bp and a depth of 20 million reads. For RNA-seq analysis, adapters and low-quality bases were removed using Trim-Galore (https://github.com/FelixKrueger/TrimGalore, accessed on 13 July 2024). The resulting trimmed reads were aligned to the human genome (hg38) or mouse genome (Mus_musculus_balbcj.BALB_cJ_v1) using the MCMV-wt viral genomes with the STAR aligner [56]. Subsequently, featureCounts [57] was utilized to count the aligned reads, generating a count matrix for downstream analyses. To quantify the MCMV gene expression, counts per million (cpm) of MCMV genes were counted using (MCMV gene reads/total reads) × 1,000,000, and the log2cpm value was calculated using log2 (cpm + 1). The expression patterns of MCMV genes were then effectively depicted via heatmaps created with the R package pheatmap (https://davetang.github.io/muse/pheatmap.html, accessed on 13 July 2024). For the mock, we split the fastq file into two halves as replicates for the analysis.

Statistics. Statistical analyses were conducted using GraphPad Prism version 9.0.2 (GraphPad Software, San Diego, CA, USA). Data are presented as the means ± standard deviation unless otherwise stated. Statistical significance was set at *p <* 0.05. A two-way analysis of variance (ANOVA) was employed to assess the significance of differences among the vectors themselves and in comparison with the wild type. A subsequent Tukey post hoc analysis was applied for correction.

## 3. Results

### 3.1. Testing the Viability and Fitness of the Large-Payload MCMV Vectors

With 101 dispensable accessory genes, an MCMV vector can theoretically accommodate a maximum payload capacity of 70 kbp (~30% of its wild-type genome). The organized genome structure enables the deletion of these genes in clusters, given that most accessory genes are located toward the genome termini (Figure 1a). Five gene clusters (I–V) that did not contain a previously identified essential gene could be identified in these assessor gene regions [26]. To exclude synergetic negative effects on viral fitness, first, a set of deletion mutants with permutations of the identified deletion clusters were generated (see Figure 1 and Table 1 for details). Previously, it was shown that a deletion mutant comprising the deletions Δm01-17+Δm144-158 (ΔI+II) yielded a replication-competent, non-attenuated vector offering approximately 23 kbp capacity in vitro [27]. To maximize the capacity, deletions of the clusters m106-109 (IV), m128-141 (V), and m159-170 (III) were additionally implemented individually or in combinations using BAC recombineering, ultimately leading to a genome lacking all five identified clusters (ΔI+II+III+IV+V) devoid of 26% of the wild-type genome size and offering ~60 kbp of payload capacity. The viability of all constructs was evaluated by transfecting permissive primary murine embryonic fibroblasts (MEFs) with purified BAC DNA derived from two independent clones in three independent experiments for each construct. The transfected cultures were observed for plaque formation daily. All tested mutants induced plaque formation after transfection (see Table 1), indicating that all genome variants that we generated were viable. Thus, any synthetic lethality among all the deleted genes could be ruled out. 

**Figure 1 vaccines-12-00791-f001:**
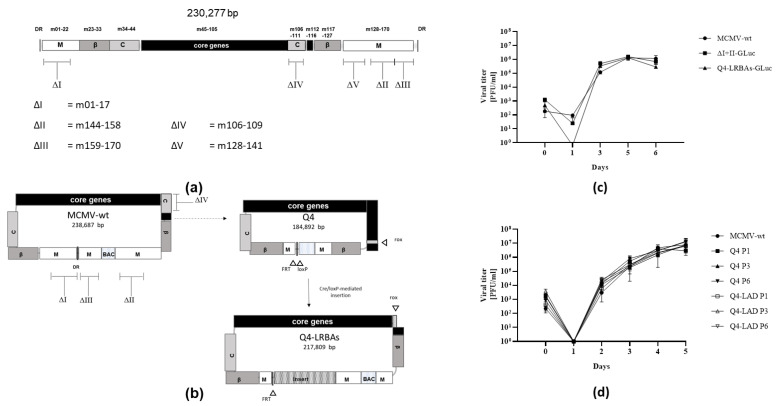
(**a**) The genomic organization of MCMVs (adapted from [26]) features distinct gene blocks. Centrally positioned are the core genes (depicted by black boxes), which exhibit high conservation across herpesvirus families. Adjacent to the core genes are the CMV-specific genes (depicted in light gray and denoted with ‘C’). Towards the termini, homologs shared among the β-herpesviruses are observed (dark gray boxes denoted with ‘β’). Large gene blocks specific to MCMVs are located at the termini (white boxes). To enhance the cargo load capacity, up to five gene blocks of accessory genes (ΔI-V) were selectively deleted. (**b**) The MCMV vector map illustrates the wild-type MCMV-BAC [47] and the BAC cassette (white box, denoted with ‘BAC’) that was inserted between the deletion clusters II and III to create a circular genome intermediate. The Q4 vector (featuring combined deletions ΔI-IV) and the indicated insertion sites for different site-specific recombinases (open triangles denoted with FRT, loxP, and rox) and an insertion mutant (Q4-LRBAs) were designed to restore the genome size nearly to that of the wild type (the stuffer is indicated by the striped box). The Q4 vector provides a cargo load capacity of almost 46 kbp. Q4-LRBAs represent loaded Q4 vectors with an insert of approximately 37 kbp. (**c**) Depicted is the result of a multistep growth analysis carried out by infecting MEFs with ΔI+II-GLuc and Q4-LRBAs-GLuc in comparison with the BAC-derived MCMV-wt Smith strain at an MOI of 0.1. Supernatants were collected on the indicated days and titrated using a plaque assay in technical triplicate. The viral titers depicted in (c) represent the results of a single representative experiment. (**d**) Depicted are multistep growth analyses testing various passages, denoted as P1, P3, and P6, for both the empty MCMV vector Q4 and the cargo-loaded vector Q4-LAD using the BAC-derived MCMV-wt control as described in (**c**).

Interestingly, however, as soon as the genome size was truncated to <82% of the wild-type genome, the appearance of detectable plaques was notably delayed, as plaque formation started 8–10 days post-transfection, which was 5–6 days later than the rescue time for the BAC encoding for the wild-type genome. In addition, a distinct plaque phenotype could be observed during the reconstitution of the ΔI+II+III+IV+V-mutant, as it resembled a comet rather than a radial-like spread of the cytopathic effect (CPE). The delay or change in plaque morphology observed for the mutants with larger deletions suggested an attenuation of viral growth or viral release due to either the cumulative negative effects of some of the deleted accessory genes or reaching a suboptimal genome size for packaging. 

Since the ΔI+II+III+IV+V-mutant showed an altered plaque morphology in addition to its attenuated phenotype, we chose MCMV-ΔI+II+III+IV as the basis for further study, as it showed only growth attenuation. Therefore, we reconstructed the deletions present in MCMV-ΔI+II+III+IV in a new BAC construct that differed in terms of site-specific recombination sites to tailor it for our recently described method that facilitates transgene insertion [49]. This modified vector was termed Q4 (Figure 1b), and, due to our genetic engineering, it possessed a theoretical payload capacity of approximately 46 kbp (~19% genome deletion). We also assessed the rescue of Q4 and observed delayed plaque formation accompanied by a wild-type-like cytopathic effect similar to that in MCMV-ΔI+II+III+IV (Table 1). 

Next, we aimed to assess the capacity of Q4 to accommodate large transgenes. The stable maintenance of transgenes is an important criterion for viral vectors but is frequently neglected. Hence, two independent stuffer DNAs were inserted into the Q4 BAC, resulting in the Q4-LAD and Q4-LRBAs BACs (Figure 2). LAD is an AT-rich DNA sequence based on an inactivated human adenovirus genome (34 kbp) and a non-coding portion of the human LRBA (LPS-responsive beige-like anchor protein) gene that is GC-rich (36 kbp). The Q4-LRBAs BAC was further modified by inserting two different transgene expression cassettes encoding for Gaussian luciferase (GLuc), resulting in Q4-LRBA-GLuc. First, we repeated the rescue assay described earlier, including the transfection of the MCMV-wt BAC as a positive control, Q4 as an attenuated control, and all the new Q4-derived BACs (Table 2).

Interestingly, after the transfection of all three recombinants, which received large DNA inserts, a retrieval of wild-type-like reconstitution (3–4 days after transfection) was observed. The wild-type-like in vitro fitness of the recombinant Q4-LRBAs-GLuc carrying the largest insert was confirmed through multistep growth analysis (Figure 1c), where we used the wild-type MCMV and the previously described lower-capacity vector, MCMV-ΔI+II-GLuc, as positive controls. Given that the genome size of these recombinants was artificially restored to nearly wild-type levels (<5%; refer to Table 2), the factor influencing the initial impaired viral fitness for both Q4 and MCMV-ΔI+II+III+IV appeared to be related to the non-optimal genome size. The loss of function of any of the deleted accessory genes alone or in combination among the regions encompassing m01-17, m106-109, and m144-170 seemed to have a minor impact. Notably, the empty Q4 vector, which reproducibly showed delayed plaque formation in the rescue assay, regained wild-type-like growth characteristics already after the first passage in MEFs (see Figure 1d), indicating fitness-driven genome adaptation to compensate for the suboptimal genome size, which most likely attenuated the viral reconstitution.

### 3.2. Viral Genome Stability

The observed increase in viral fitness prompted an inquiry into the integrity of the genome of our MCMV constructs. To this end, we isolated DNA from sucrose cushion-purified cell-free viral particles harvested from different passages (P1, P5, P10, etc.) by sub-culturing lysates of three independent reconstitutions (suffixes A, B, and C) of MCMV-wt, Q4, Q4-LRBAs-GLuc, and Q4-LAD. The genomes were assessed by using Illumina next-generation sequencing. The substantial long-range changes in read coverage obtained after the paired-end reads were mapped to the reference genome allowed us to identify gene deletions, multiplications, and bigger rearrangements (see Figure 3a–d). The MCMV-wt showed a consistent read frequency distribution across the genome and among passages, indicating genomic integrity. However, the loss of coverage with the BAC cassette region (nt 178.167–184.612) confirmed our approach to the detection of the loss of genetic material with this simple analysis. This sequence was not sustained beyond passage 5, as expected, resulting in complete loss (Figure 3a), as expected. The removal of this extra artificial sequence of approximately 6 kbp, which exceeded the viral genome size, was previously reported through Southern blotting upon passaging [47]. The MCMV-wt genome appeared to show some instability starting at passage 15. It was not yet further addressed and was assumed to display the effect of cell culture adaptation upon multiple passages, an effect that is known for many viruses. The two prominent sharp peaks, one of which was located in the BAC region, while the other was located in the enhancer sequence for the *ie1* gene (nt 183.251–183.850), were present in all analyzed samples that carried the corresponding sequences and were considered as sequencing or mapping artifacts. In passage 1, the empty MCMV vector Q4 exhibited a heterogeneous coverage plot (Figure 3b). However, the read-frequency noise transformed into a defined pattern by passage 5 and remained stable in subsequent passages. In this pattern, some loci appeared to have increased coverage, hence indicating the amplification of the respective viral sequences. Intriguingly, the affected regions were different for independent reconstitutions (compare Figure 3b with Appendix A), indicating a low number of random events for fitting the genome size, which was most likely very soon after or upon viral rescue. The assay used could not detect host gene contributions, as it mapped only viral reads. Nonetheless, there was an initial genome instability in Q4, which resulted in amplifications of genome regions. Thus, a set of different genomes appeared soon after reconstitution, and they may have gained extra sequences that most likely filled up the wild-type genome, resulting in one or a few genome forms being predominant in the later passages. DNA replication intermediates potentially obscuring the data were ruled out by using purified cell-free viruses as samples. Noteworthily, unlike MCMV-wt, Q4 stably maintained the BAC cassette since the smaller genome could most likely accommodate the extra DNA. 

Upon transfection, the Q4-LRBAs-GLuc BAC (~4% undersized) was reconstituting as a wild-type construct. Here, the coverage was evenly distributed during passaging, and no significant locus loss could be mapped (Figure 3c and Appendix A). Moreover, Q4-LAD (~5% undersized) also maintained its genomic integrity over the course of 10 passages (Figure 3d and Appendix A). Remarkably, none of these reduced-size constructs experienced loss of their BAC cassette, unlike the initially oversized MCMV-BAC, which carried the wild-type genome. This observation suggests that the enhanced stability of the loaded Q4 vectors was achieved through the incorporation of the large artificial inserts. The presence of these insertions appeared to compensate for the suboptimal genome size of the empty vector, eliminating the need for random recombination events to restore its fitness.

### 3.3. MCMV Vectors Showed a Broad Host Range

The potential translational application of MCMVs highly depends on their ability to cross the species barrier upon transgene delivery. Thus, we tested the cross-species host range of our MCMV vector. To this end, a set of animal and human cells were infected using MCMV vectors (ΔI+II-GLuc and Q4-LRBAs-GLuc) expressing the secreted reporter gene GLuc, which allowed the assessment of the enzymatic activity in the supernatant of the infected cultures over a course of up to 5 dpi (measured on days 1, 2, and 5). We set up assays targeting mouse (MEF, murine embryonic fibroblast), human (HFF, human foreskin fibroblast), ARPE-19 (human retinal pigment epithelial cell line), A549 (human adenocarcinomic alveolar basal epithelial cell line), 293A (human embryonic kidney cell line), 911 (human retinal pigment epithelial cell line), HeLa (human epithelioid tumor cell), primate CV-1 (African green monkey kidney fibroblast cell line), Vero E6 (green monkey kidney fibroblast cell line), bovine MDBK (bovine kidney cell line), porcine VR1BL (fetal porcine retina cell line), PK-1 (porcine kidney cell line), canine MDCK (canine kidney cell line), bat EpoNi (renal epithelium cell line), chicken DF-1 (chicken fibroblast cell line), and LMH (chicken hepatocellular carcinoma cell line) cells, which were transduced using ΔI+II-GLuc or Q4-LRBAs-GLuc MCMV vectors with an MOI of 3 (Figure 4a–c). To our surprise, the MCMV vectors demonstrated a broad host spectrum with comparable transgene expression levels to those of permissive MEF cells, except for MDCK cells, for which an increase in GLuc activity was not measurable. Notably, no significant difference was observed between the large deletion vector (Q4-LRBAs-GLuc) and the published vector (ΔI+II-GLuc) across all tested cells and time points. The expression kinetics, which were assessed in fibroblasts from the naturally permissive host of MCMV, exhibited a high expression at day 1, followed by an increase on the subsequent day and a stable expression level thereafter. Given the high MOI in MEFs, cell lysis occurred at later time points. The prolonged half-life of the luciferase enzyme accounted for the static relative luminescence unit (RLU) values observed between day 2 and day 5. Most tested cells showed similar kinetic profiles, with some exhibiting higher expression (911, Vero E6, MDBK) or slightly lower levels (HFF, A549, 293A). CV-1 and ARPE-19 displayed very similar expression kinetics, with a notable reduction in RLU at day 5. In the HeLa, VR1BL, PK-15, and LMH cell lines, the GLuc expression kinetics were delayed, but VR1BL reached the levels seen in MEFs on day 5. Conversely, EpoNi and DF-1 exhibited stable transgene expression levels throughout the experimental duration. Moreover, both vectors maintained similar transgene expression levels and durations, testifying that the accessory genes that were deleted did not drastically change either the viral host range or the cytotoxicity.

### 3.4. Kinetics of DNA Replication in MCMV-Infected Cells 

The persistent and reliable expression of transgenes observed with MCMV vectors in a diverse array of cells beyond their natural hosts defied conventional expectations. To explore the possibility that MCMV vectors could progress through later stages of their viral cycle beyond viral entry, we conducted an analysis of viral DNA replication in a set of cells infected with a high MOI of both vectors, ΔI+II-GLuc and Q4-LRBAs-GLuc (Figure 4d,e). As anticipated, there were no differences in the trends between the two vectors in terms of DNA replication in MEFs at any time point, starting with approximately ten viral DNA copies/cell, increasing in the late stage, reaching a peak at 3 dpi, and staying on a plateau. Unlike in MEFs, the viral genome copies were lower in all cell lines tested at all time points (ARPE-19, A549, 293A, 911, and Vero E6). Nevertheless, the increasing genome copy number over time (until day 5 for 293A and Vero E6) was evident in all tested cells and generally slightly higher upon infection with the Q4-based vector. Interestingly, the viral genome copy number in ARPE-19 apparently increased from 14 hpi to 36 hpi, and the DNA copy number was maintained in the following days. In principle, this was in accordance with the luciferase transgene expression. This indicated that the MCMV-based vectors could indeed establish at least an abortive infection that resulted in long-lasting DNA replication in cells of non-host origin in vitro.

### 3.5. Quantification of Viral Particle Release from Infected Non-Rodent Cells 

This observation of genome amplification prompted us to test the possibility of the formation and release of infectious particles in these cells using a multistep growth curve. A selected set of cells (MEF, HFF, ARPE-19, A549, 293A, 911, CV-1, Vero E6, MDBK, and DF-1) were infected with MCMV-wt, ΔI+II-GLuc, or Q4-LRBAs-GLuc. The cell-free viruses were harvested at designated time points (days 1, 3, 5, and 7), and the newly produced infectious particles were determined via a plaque assay performed on fully permissive MEFs (Figure 5). As already indicated (compared with Figure 1c), all three MCMVs grew with comparable kinetics in MEF cells, reaching a plateau of roughly 10^5^ PFU/mL of infectious particles at 3 dpi. As expected, the infectious particle release rate was unmatched by any other cell line, yet, very surprisingly, it was only reduced by 10–100-fold in the cases of 911 and Vero E6 cells. Time-dependent particle release was also measured in 293A, ARPE-19, CV-1, and, to some extent, HFF cells. In terms of kinetics, there were no differences among the tested viruses, but it was noteworthy that the MCMV vectors yielded more viral particles in Vero E6 and 911 than in the wild type. We also noticed that the deletion viruses generally produced more infectious particles in 911 and Vero E6 cells than MCMV-wt did.

In contrast to the observable increase in DNA copies and the robust transgene expression, no quantifiable infectious particle release was detectable in A549 cells; likewise, the transducible DF-1 and MDBK cells appeared to be non-permissive. 

### 3.6. Transcriptomes of the MCMV and Large MCMV Cargo Vector in Human and Mouse Cells

The obtained data prompted us to analyze the early and late viral transcriptomes of the MCMVs (MCMV-wt, Q4-LRBAs-GLuc) in MEFs, ARPE-19, A549, and 911 (Figure 6). As expected, both viruses showed no difference in the transcriptome in MEFs, neglecting the deleted genes and the luciferase. However, the overall gene expression levels of the Q4-based vector appeared to be higher at 31 dpi, which was in agreement with the difference observed in the DNA copies in favor of Q4-LRBAs-Gluc (see Figure 4e). There was not an apparent difference in the kinetics between human-derived cells and MEFs. Noteworthily, there were some genes, such as M45, M37, and M38, that were overexpressed in all three human cell lines compared with the MEFs. Intriguingly, all three gene products are known virulence factors that exhibit an anti-apoptotic role, among other functions [58,59,60]. In addition, in 911 cells, a generally high level of IE transcripts was measured. Nevertheless, the variations observed in viral transcription among the different human cell lines cannot explain the significant differences in permissiveness. This suggests that host factors influencing later stages of the viral cycle may play a pivotal role in determining the observed disparities in permissivity in cross-species settings.

## 4. Discussion

Cytomegalovirus-based vectors have been shown to be excellent vaccine vectors when applied to the same species. For example, rhesus CMV vectors induced promising protection in primate models when applied to the same species [3,18,19,20,32]. One characteristic feature that all CMV vectors possess—even spread-deficient ones—is the ability to evoke a robust circulating T cell effector memory response [1,2,16,61]. Typically, MCMVs serve as a well-established animal model for their human counterparts. The cross-species application of MCMVs is certainly inhibited by the observation that CMVs appear to be highly species-specific in certain experimental settings. The blocking of viral replication seems to happen after the penetration of cells [62]. Brune and colleagues identified two challenges that MCMVs must overcome in human cells: (i) apoptosis and (ii) antiviral immune response [63]. In our investigation, we confirmed the successful delivery and even long-lasting expression of transgenes using MCMV vectors in fibroblasts and epithelial cells derived from various species, including mice, humans, primates, pigs, cows, bats, and even chickens. Here, Vero E6, 911, and 293A cells demonstrated the ability to support infectious particle formation, rendering them semi-permissive; nevertheless, other cells did not seem to have this ability. A kinetic viral transcriptome analysis did not allow the identification of characteristics that explained why viral growth was supported or not in human cells, since there were no remarkable differences from fully permissive MEFs. Noteworthy, in a set of cells supporting MCMV replication, all of them demonstrated impairment in either (i) apoptosis (293A, 911) or (ii) antiviral response (293A, 911, Vero E6). Vero E6 cells are already known to support many viral infections, which is certainly due to their inherent defects in interferon production and in the antiviral RNAi pathway [64]. In 293A and 911, the apoptosis and interferon signaling pathways were impaired as well. In both cell lines, three adenoviral genes, *E1a*, *E1b*, and *pIX,* are expressed. Here, E1a was identified to reduce interferon-alpha and -beta expression by interacting with TMEM173/STING and, consequently, influencing the cGas-STING pathway [65]. Further, *E1a* interfered with the E2F1–RB interaction [66]. The consequent accumulation of p53 and the inevitable apoptosis are counteracted by *E1b* gene products (mainly 55K), which possess an anti-apoptotic function [67]. The impairment of innate immunity can undoubtedly support viral replication upon viral entry. In the context of MCMVs, it was shown that the cGAS/STING axis plays a critical role in early antiviral defense mechanisms. Nevertheless, while cGAS KO and STING KO mice did not show increased susceptibility to lethal MCMV infection, mice lacking TLR and RLR signaling exhibited enhanced vulnerability [68]. Noteworthy, it has already been shown that MCMVs can cross the species barrier when interfering with either the apoptosis or E2F pathways [63,69,70]. Interestingly, 911 cells are more permissive than 293A cells for MCMVs, which cannot be explained by the presence of helper functions for transforming adenovirus genes. These data, together with our transcriptome analysis, which showed few differences in viral gene expression among these cells, indicate that yet-unknown host factors that most likely act at later stages of the viral cycle are involved in the regulation of host restriction or the permissivity for MCMVs. It is worth noting, particularly regarding both tested vectors with large deletions, that the permissivity of 911 cells matched the permissivity of MEFs, which are used as a standard for the propagation of MCMVs in vitro. In contrast, ARPE-19 cells, which are the closest available match for the precursor of 911 cells [40], are semi-permissive at best. This indicates that cell transformation by oncogenes (such as *E1a* and *E1b* in the present case) can render human primary cells permissive to MCMV replication. This suggests that the oncolytic potential of MCMV vectors, especially the recombinant versions ΔI+II and the Q4-LRBAs-based ones presented here, should be evaluated. Here, the immunological features of MCMVs remain especially elusive in a cross-species setting. Promising results were obtained for an RhCMV vector applied to cynomolgus macaques [32]. Notably, the deletion viruses generally produced more infectious particles in 911 and Vero E6 cells than MCMV-wt did. We can only speculate that either the constructs with the large deletions may have lacked some accessory genes, which prevented cell lysis, or we deleted some accessory genes, which rendered the wild-type virus more cell-associated in some cell types (MCK-2, for example); therefore, we detected somewhat more released viruses for the deletion mutant in some cell types.

Interestingly, we observed pronounced transgene expression in A549, Vero E6, and MDBK after transduction with MCMV vectors. From this group, A549 was further analyzed regarding the genome replication of the vector and total transcriptome. We could not identify any significant differences in viral transcription or DNA replication in comparison with the other cross-species hosts (e.g., 911) that released infectious particles. Therefore, we believe that in A549 cells, there must be a late blocking of viral release (e.g., nuclear egress or secondary envelopment) that does not affect viral gene expression.

In this study, we also wished to challenge the applicability of MCMVs in terms of their payload capacity. Here, the organized genome structure conveyed the impression of deleting the long terminal accessory gene blocks—adding up to 101 genes—to generate a theoretical capacity of 70 kbp. Interestingly, the deletion of 61 accessory genes (=60 kbp) resulted in an in vitro viable mutant. However, the vector (Q4) introduced here offered a payload capacity of 46 kbp, translating into an approximately 19% deletion of its genome with three site-specific recombination sites for the one-step integration of any insert. An analysis of this deletion mutant revealed profound genomic instability, leading to initially reduced viral fitness that was naturally cured through gene duplication or artificially cured through the insertion of stuffer DNA. 

A potential explanation for this observation was attributed to the viral metastability that describes the ambivalent demands of a fragile capsid that simultaneously withstands a strong inner pressure [71]. As herpesviruses replicate their genome in the form of concatemers, terminase enzymes cleave them at specific positions after insertion through the portal [72]. This process requires the genome to meet a certain threshold size [73]. A coping mechanism of rapid gene expansion is known for many organisms [74,75,76,77,78,79,80] and has even been documented in dsDNA viruses and pox viruses [81]. In addition, CMVs, RhCMVs, and HCMVs can undergo drastic and rapid genome rearrangement when adapting to a cell culture system [82,83]. Further, it has been reported that a deletion mutant of guinea pig CMV (GPCMV) compensates for its suboptimal genome size by duplicating its repeats [84]. The stable maintenance of the inserts for at least 10 passages in vitro, which is feasible for highly effective vector production, is encouraging for further vector application. To date, it remains elusive if smaller-payload MCMV vectors, namely, ΔI+II-MCMV, represent a suboptimal genome size that is also subject to the same genome size adaptation process seen in the case of the empty Q4 vector. Furthermore, similar processes could affect representative vectors from other families; therefore, the prevention of undesired genome adaptations in viral vectors might require a payload design to ultimately add up to a genome size that is approximately that of the wild type.

An important limitation of this study is that all the data presented here were solely derived from in vitro assays and must be confirmed in in vivo models. The context of in vivo infection involves the infection of multiple cell types and induces innate and adaptive immune responses, which can critically influence the genomic stability, vector tropism, and gene expression presented here.

## 5. Conclusions

In conclusion, it was possible to generate an MCMV-based vector that stably accommodated a 35 kbp payload that propagated to wild-type-like titers and did not show attenuation in vitro compared with the MCMV-wt. The new vector platform characterized in this study, Q4, had a cloning capacity of 46 kbp for transgenes, and it can be inserted into three independently accessible sites. Additionally, MCMVs proved to be a suitable vector for potential administration in veterinary or human medicine since MCMV-mediated gene delivery appeared to have a very broad host range. All of the features tested in this study support the idea that MCMVs provide a promising cross-species replication-competent vector for the development of new vaccines with unparalleled capacity for many different antigens or multiple variants of the same antigen. Since progressive MCMV replication in human cell lines was evident, we believe that it is worth investigating the oncolytic potential of MCMV-based vectors. 

## Figures and Tables

**Figure 2 vaccines-12-00791-f002:**
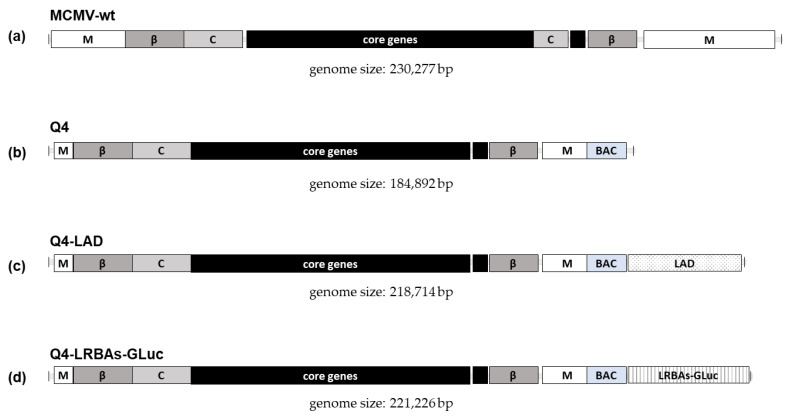
Scale representation of four MCMV genomes characterized in detail in this study. Each genome is represented to scale, illustrating the relative sizes and modifications between the wild-type and the engineered variants. For feature explanation, refer to Figure 1. (**a**) Wild-type MCMV genome (MCMV-wt), comprising 230,277 base pairs (bp). (**b**) Q4 vector, a deletion variant of the MCMV genome, reduced to 184,892 bp by the removal of gene blocks ΔI-ΔIV. (**c**) Q4-LAD vector, derived from the Q4 vector, incorporating a large insertion termed LAD, resulting in a total genome size of 218,714 bp. (**d**) Q4-LRBAs-GLuc vector, similar to Q4-LAD but containing the LRBAs-GLuc insertion instead, with a total genome size of 221,226 bp.

**Figure 3 vaccines-12-00791-f003:**
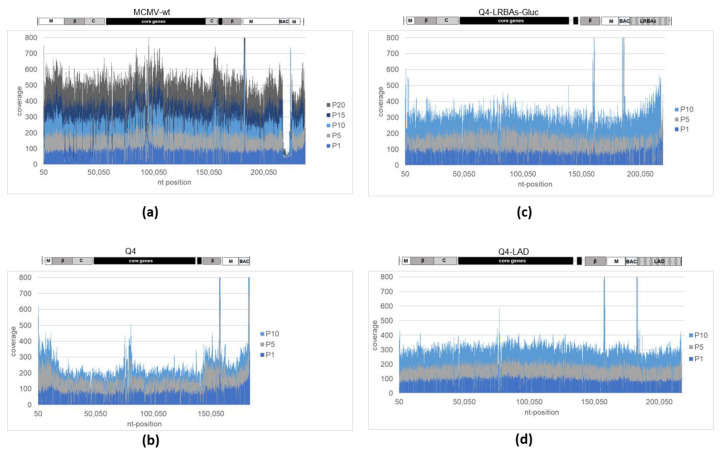
Illumina paired-end sequencing was employed to assess the read coverage of MCMV vectors. DNA extracted from cell-free viruses at designated passages on mouse embryonic fibroblasts (MEFs) served as the sequencing sample. Reads were aligned to reference genomes (see Appendix A for the GenBank accession numbers), and deletions were further analyzed for the frequency of deletion reads. Panel (**a**) illustrates the read coverage of the MCMV-wt genome across various passages. Panels (**b**–**d**) represent the read coverage for the empty Q4 vector, Q4-LRBAs-GLuc, and Q4-LAD, respectively.

**Figure 4 vaccines-12-00791-f004:**
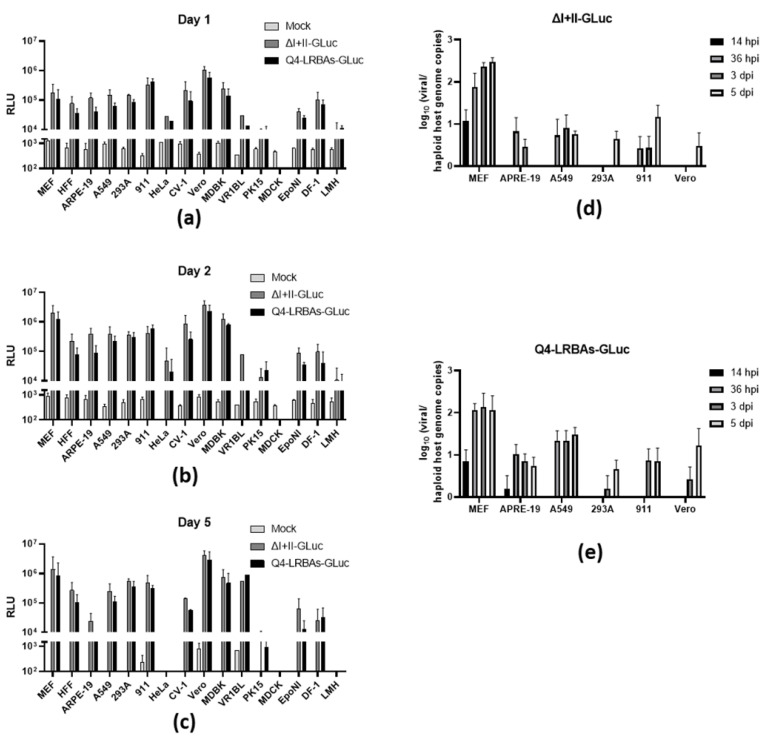
In vitro transduction efficiency of MCMV vectors in cell lines derived from different species. In panels (**a**–**c**), the Gaussian luciferase activity in the supernatant of cultures induced by treatment with the indicated MCMV vectors at an MOI of 3 at 1, 2, and 5 days post-infection (dpi), respectively, is depicted. The luciferase activity was measured indirectly during substrate conversion and is presented here in relative light units (RLUs). Among the transduced cells are fibroblasts of murine, human, and chicken origin, as well as epithelial cells of human, primate, bovine, porcine, canine, bat, and chicken origin. “Mock” represents data from equally treated cells despite viral infection. Data were normalized to the values measured 1 h post-infection (hpi). The values shown are the means of three independent experiments measured in technical triplicate, and the error bars indicate standard deviations. As shown in panels (**d**,**e**), the viral genome copies were determined by using M45 gene-specific qPCR at different times after infection of the indicated cell lines with the indicated MCMV vectors. The total DNA extracted from infected MEF, ARPE-19, A549, 293A, 911, and Vero E6 cells was assessed at four different time points (14 hpi, 36 hpi, 3 dpi, and 5 dpi). The resulting viral copy numbers were calculated per haploid genome count, which was determined by using the control qPCR to amplify the GAPDH genes of the respective hosts. (**d**) The data from three independent experiments are shown pooled together for the MCMV vector ΔI+II-GLuc. (**e**) Same as (**d**) for Q4-LRBAs-GLuc.

**Figure 5 vaccines-12-00791-f005:**
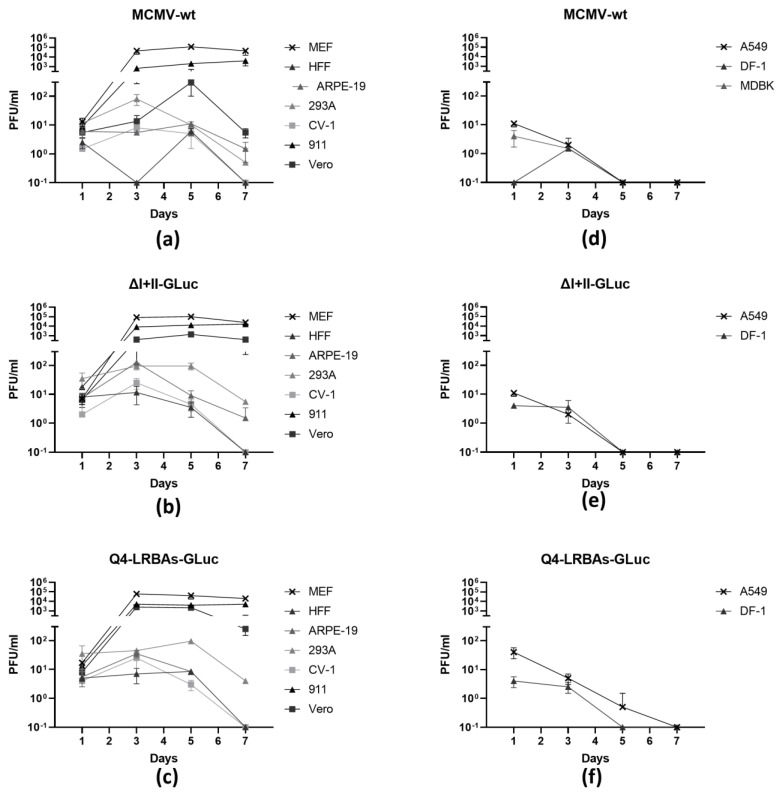
Release of infectious particles into the supernatant upon infection of various cells with MCMV–wt (**a**,**d**) and the MCMV vectors ΔI+II~–GLuc (**b**,**e**) and Q4–LRBAs–GLuc (**c**,**f**). The indicated cells were infected with the respective viruses at a multiplicity of infection (MOI) of 3 and washed extensively, and the potentially de novo-generated virions were quantified using a standard plaque assay on murine embryonic fibroblasts (MEFs). In the left panels (**a**–**c**), cells that produced a detectable titer at 5 dpi or later are shown. In the right panels (**d**–**f**), cells that did not produce infectious virions at 5 days post-infection (dpi) or later are depicted. The titers were particularly high in 911 and Vero E6 cells, in addition to the permissive MEFs. The graphs show the results of data from three independent experiments.

**Figure 6 vaccines-12-00791-f006:**
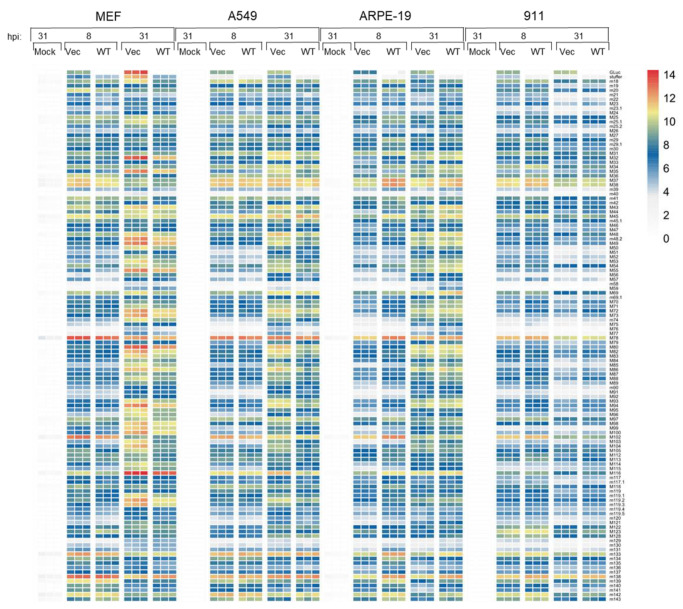
Heatmaps depicting viral mRNA transcription in MEF, A549, ARPE-19, and 911 cells following infection with either Q4-LRBAs-GLuc (Vec) or wild-type MCMV (WT) at early (8 h post-infection (hpi)) and late (31 hpi) time points. The log2 of normalized read counts by counts per million (cpm) is presented as the MCMV transcript level. Each column corresponds to an individual experiment (mock experiments are split into two replicates). The genes coding for Q4-LRBAs-GLuc are presented. The expression of each gene has been scaled. Samples with relatively high expression of an MCMV gene are marked in red, and samples with relatively low expression are marked in yellow and blue. White areas represent genes with no counts.

**Table 1 vaccines-12-00791-t001:** The genome size and rescue efficiency of the deletion mutants of MCMV used in this study.

MCMV Vectors	BAC Size [bp]	Predicted Viral Genome Size [bp]	Payload Capacity [bp] *	Payload Capacity [%] ***	Plaque Formation [day]
MCMV-wt	238,687	230,277	n.a. **	n.a.	3–4
ΔIII	226,758	226,758	−3519	1.53	3–4
ΔIV	231,515	231,515	1238	−0.54	3–4
ΔV	225,396	225,396	−4881	2.12	3–4
ΔI+II	207,594	207,594	−22,683	9.85	3
ΔIII+IV	219,586	219,586	−10,691	4.64	3–4
ΔIII+V	213,467	213,467	−16,810	7.30	3–4
ΔI+II+III	195,666	195,666	−34,611	15.03	3–4
ΔI+II+V	194,303	194,303	−35,974	15.62	3–4
ΔI+II+IV	200,422	200,422	−29,855	12.96	3–4
ΔI+II+III+IV	188,494	188,494	−41,783	18.14	8–9
ΔI+II+IV+V	182,375	182,375	−47,902	20.80	8–9
Q4	184,892	184,892	−45,385	19.71	8–9
ΔI+II+III+IV+V	170,000	170,000	−60,277	26.18	10

* Predicted viral genome size of MCMV-wt deducted from the BAC size; ** not applicable; *** proportional to the wild-type virus size.

**Table 2 vaccines-12-00791-t002:** The genome size and rescue efficiency of the MCMV vectors with large inserts.

MCMV Vectors	BAC Size [bp]	Predicted Viral Genome Size [bp]	Payload Capacity [bp] *	Payload Capacity [%] ***	Plaque Formation [day]	Insert Size [bp]
MCMV-wt	238,687	230,277	n.a. **	n.a.	3–4	N/A
Q4	184,892	184,892	−45,385	19.71	8–10	N/A
Q4-LAD	218,714	218,714	−11,563	5.02	3–4	33,822
Q4-LRBAs-GLuc	221,266	221,266	−9011	3.91	3–4	36,374

* Predicted viral genome size of MCMV-wt deducted from the BAC size; ** not applicable; *** proportional to the wild-type virus size.

## Data Availability

The sequences of the newly constructed plasmids and bacmids have been submitted to GenBank, and their accession numbers are listed in Appendix A, respectively. The NGS data used in this study are available online at https://doi.org/10.5281/zenodo.12736850 accessed on 13 July 2024, for the genome stability analysis; https://doi.org/10.5281/zenodo.12740203 accessed on 13 July 2024, for the transcriptome analysis of the infected MEFs; https://doi.org/10.5281/zenodo.12740251 accessed on 13 July 2024 for the transcriptome analysis of the infected A549; https://doi.org/10.5281/zenodo.12740285 accessed on 13 July 2024, for the transcriptome analysis of the infected 911; and https://doi.org/10.5281/zenodo.12740315 accessed on 13 July 2024, for the transcriptome analysis of the infected ARPE-19.

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
