# Peer review of "Construction and Characterization of a High-Capacity Replication-Competent Murine Cytomegalovirus Vector for Gene Delivery"

_vaccines, 2024, doi:10.3390/vaccines12070791_

Round 1

Reviewer 1 Report

Comments and Suggestions for Authors

1. The researchers used BAC technology to engineer replication-competent recombinant MCMVs with deletions of up to 26% of the wild type genome. This vector platform allows for the delivery of large genomic DNA segments, up to 35 kbp, along with reporter genes.

2. The study found that the insertion of diverse foreign DNAs into the MCMV vector alleviated the delayed plaque formation phenotype and remained stable through serial in vitro passages. 

3. The MCMV vectors demonstrated a broad host spectrum, successfully transducing not only mouse cell lines but also non-rodent mammalian cells, including those of human, monkey, bovine, and bat origin. 

This study presents a novel MCMV vector platform with a high cloning capacity and broad host range, providing new opportunities for gene delivery and vaccine development.

Author Response

Comments 1. The researchers used BAC technology to engineer replication-competent recombinant MCMVs with deletions of up to 26% of the wild type genome. This vector platform allows for the delivery of large genomic DNA segments, up to 35 kbp, along with reporter genes.

Comments 2. The study found that the insertion of diverse foreign DNAs into the MCMV vector alleviated the delayed plaque formation phenotype and remained stable through serial in vitro passages. 

Comments 3. The MCMV vectors demonstrated a broad host spectrum, successfully transducing not only mouse cell lines but also non-rodent mammalian cells, including those of human, monkey, bovine, and bat origin. 

Response 1-3. We thank Reviewer 1 for the kind reviewing of our manuscript. Indeed, you commented on our main conclusions correctly. We report on the first replication competent high capacity MCMV vector, which, based on our data, is ready for cross-species applications as recombinant vaccine platform. We also tested the stability of our vectors, which frequently overlooked in this stage of research, but we believe that it is an important issue, that has to be addressed in the earliest possible stage of development.

Reviewer 2 Report

Comments and Suggestions for Authors

The manuscript "Construction and Characterization of a Replication Competent High-Capacity Murine Cytomegalovirus Vector for Gene Delivery" from André Riedl and colleagues reports the construction and verification of a series of new MCMV vectors and platform. Using BAC technology and targeting five gene blocks of MCMV, the authors constructed replication-competent recombinant MCMVs and introduced large genomic DNA payload, up to 35 kbp. Moreover,MCMV vectors could transduce not only mouse cell lines but also non-rodent mammalian cells appeared to have a broad host range, and large inserts of foreign genes remained stable through in vitro passages. The manuscript provides a large capacity and cross-species applicable MCMV delivery vector platform with potential application value for basic research, vaccine development and gene therapy. Regarding the content of the article, there are minor errors in the article, which requires detailed revisions.

Tabel 2 has mentioned genomic size data for different MCMV vectors. Can you provide genome electrophoresis results for MCMV wt, Q4, Q4-LAD, and Q4-LRBAs GLuc to compare their size differences more intuitively?

What possible reasons led to the phenomenon that the deletion viruses produced generally more infectious particles in 911 and Vero E6 cells than MCMV-wt did?

Fig.5 (d)-(f) shows there was no quantifiable infectious particle release detectable in A549, DF-1 and MDBK cells, but robust transgene expression was observed in these cells in Fig.4, What is the reason for the mismatch between these two results?

The verification results of this manuscript are all at the in vitro cell-level. Subsequent testing of the gene delivery effect of MCMV vectors requires animal experiments. Have relevant experiments been conducted? 

Minor concerns:

Line 33: In the sentence of Several platforms for recombinant vaccines exist that offer potent neutralizing antibody response against prevalent targets, is the statement 'exist that' incorrect?

Line 36: In the sentence of “...induction of anti-tumor immunity remained to be met...”, Why is there an expression of anti tumor immunity?

Line 70: Please note that " (see also in Error! Reference source not found. a)" needs to be deleted.

Line 210: The p value should be written in italics.

Is there no annotation information in Table 1 and Table 2?

Lines 219 & 223 & 273 & 283 & 296 & 306 & 317 & 322 & 331 & 336 & 349 & 351 & 382 &  & 406 & 445 & 470...: There are many incorrect statements of “see Error! Reference source not found”. Please modify them.

The resolution of Fig.1(b) and Fig.2 is insufficient.

Line 283: should “Q$-LAD” be “Q4-LAD”?

Line 378: Please confirm if the names of the following cell lines are correct: primate (CV-1 (green monkey kidney fibroblast cell line)?  bovine (MDBK (bovine kidney cell line))? porcine (VR1BL (fetal porcine retina cell line)?  canine (MDCK (canine kidney cell line)), bat (EpoNI (renal epithelium cell line)) and chicken (DF-1 (chicken fibroblast cell line)?

Line 447: In the sentence of “...reaching the plateau 3 dpi of roughly 105/ml infectious particles”, 105/ml should be 105 PFU/mL.

Fig.4: (a) (b) (c) (d)... are not marked in Figure 4.

Comments on the Quality of English Language

There are some sentences in the article that are not expressed accurately enough. Please improve and edit English language to make it easier for readers to read.

Author Response

We thank Reviewer 2 for the thorough review of our manuscript. Indeed, we agree that all points you mentioned are of importance and dealing with them improved our manuscript. We are also very grateful for the in-depth proofreading.

Please, find our responses to the raised questions points by points below.

Comment 1) Tabel 2 has mentioned genomic size data for different MCMV vectors. Can you provide genome electrophoresis results for MCMV wt, Q4, Q4-LAD, and Q4-LRBAs GLuc to compare their size differences more intuitively?

Response 1) We agree that for smaller sized constructs well-chosen restriction endonucleases can produce restriction patterns, which can informatively navigate through multiple deletions. However, MCMV BACs in this study are 238-170 kbp in size. Electrophoretograms could show restriction length polymorphism with at least 12-16 bands/lane in the range maximum 14-1,5 kbp, with an equal distribution in the ideal case representing about maximum 100-110 kbp. The rest of the bands will not be resolved. This means that we can only represent about the half of the genomes even in an ideal case. Since our deletions/insertions are in different regions, it is almost impossible to find a single restriction endonuclease, which generate signature fragments representing all of our changes. If we use a specific restriction for each changes, we may have the same complexity of the message. But, we agree that all structural information is not coming through by our very complex tables. Therefore, we inserted a new figure (Figure 2) with the linear representation of the genomes shown in Table 2.

Comment 2 “What possible reasons led to the phenomenon that the deletion viruses produced generally more infectious particles in 911 and Vero E6 cells than MCMV-wt did?”

Response 2) We cannot clearly answer this question. We can only speculate and imagine two main possibilities. I) The constructs with the large deletions may lack some accessory genes, which prevent cell lysis. We can refer to that cell culture adaptation of CMV sometimes lead to deletion of anti-apoptotic viral genes, for example the loss of UL36 in the 68-1 strain, which lead to better cell lysis and more efficient release of virus particles. In this study we wanted to be sure, that we do not detect remnants of the inoculum, when testing the permissively and therefore we restricted to test the released virus particles and not the total virus production of cell associated and released particles. II) The other possibility that we deleted some accessory genes, which render the wild type virus more cell associated in some cell types (MCK-2, for example) and therefore we detect somewhat more released virus for the deletion mutant in some cell types. Nevertheless, we agree with you that it is an important point and included these aspects in our discussion (see lanes 564-570).

Comment 3) “Fig.5 (d)-(f) shows there was no quantifiable infectious particle release detectable in A549, DF-1 and MDBK cells, but robust transgene expression was observed in these cells in Fig.4, What is the reason for the mismatch between these two results?”

Indeed, we could see a pronounced transgene expression in A549, Vero E6 and MDBK using gaussian luciferase over the course of 5 days. Here, the data suggests that the expression is matching if not excelling the one of the permissive host MEF. Especially, A549 was analyzed regarding DNA replication and total transcriptome. Here, we could not identify a significant difference in transcription and DNA replication regarding e.g., 911 which released infectious particle. Here, the RNAseq data did not reveal an obvious restriction factor from the host to explain the discrepancy of moderate/high level of viral DNA-replication/transcription and the low particle outcome. Therefore, we believe that in A549 cell there must be a late block of virus release, which does not affect viral gene expression. Of course, it is interesting what mechanism contributes to diminish completely virus particle release in A549 (or MDBK, DF-1 cells) and maybe a good subject of further research. Again, thanks for pointing this important aspect; we inserted a paragraph discussing this phenomenon (see lines 571-578).

Comment 4) The verification results of this manuscript are all at the in vitro cell-level. Subsequent testing of the gene delivery effect of MCMV vectors requires animal experiments. Have relevant experiments been conducted?

Indeed, all presented data are in vitro. There were not yet any animal experiments performed, however, certainly it is our ambition to test our vectors in vivo in cross-species settings. However, organizing and carrying out animal experiments, which utilizes models other than rodents are expensive and ethically more demanding and we believe that different models will lead to different conclusions and a number of cross-species application should be tested. Therefore, we believed that it is beneficial for the further development of our MCMV-platform to share our data with the community of vaccine researchers in this early stage. We included a statement pointing of this important limitation of our presented study in the discussion (see lines 607-611).

Minor concerns:

Comment 5) Line 33: In the sentence of “Several platforms for recombinant vaccines exist that offer potent neutralizing antibody response against prevalent targets”, is the statement 'exist that' incorrect?

We modified the sentence. The new version can be found at lines 33-34.

Comment 6) Line 36: In the sentence of “...induction of anti-tumor immunity remained to be met...”, Why is there an expression of anti tumor immunity?

 Corrected to “anti-tumor response”. The corrected sentence can be found at lanes 34-37.

Comment 7) Line 70: Please note that " (see also in Error! Reference source not found. a)" needs to be deleted.

Response 7) We do not know what happened, the field codes for the figures and tables got corrupted. We tried to cure all error-messages and the field code problem too. Hopefully it worked, at least our PDF version looks OK.

Comment 8) Line 210: The p value should be written in italics.

Response 8) See the correction at lane 216.

Comment 9) Is there no annotation information in Table 1 and Table 2?

Response 9) We annotated the tables.

Comment 10) Lines 219 & 223 & 273 & 283 & 296 & 306 & 317 & 322 & 331 & 336 & 349 & 351 & 382 &  & 406 & 445 & 470...: There are many incorrect statements of “see Error! Reference source not found”. Please modify them.

Response 10) Please, see Response 7, we tried to repair it.

Comment 11) The resolution of Fig.1(b) and Fig.2 is insufficient.

Response 11) The Figure 1 b was rearranged to increase the readability. As well as in formal Figure 2 (now Figure 3, see Response 1), the font size was changed to increase the readability.

Comment 12) Line 283: should “Q$-LAD” be “Q4-LAD”?

Response 12) Certainly, it must be Q4-LAD and is now corrected accordingly (see lane 291)

Comment 13) Line 378: Please confirm if the names of the following cell lines are correct: primate (CV-1 (green monkey kidney fibroblast cell line)?  bovine (MDBK (bovine kidney cell line))? porcine (VR1BL (fetal porcine retina cell line)?  canine (MDCK (canine kidney cell line)), bat (EpoNI (renal epithelium cell line)) and chicken (DF-1 (chicken fibroblast cell line)?

Response 13) The cell lines listed in lines 392-401 were mostly correctly named with the exception of EpoNi, which was misspelled.

1) Primate (CV-1): The CV-1 cell line is derived from the kidney fibroblasts of an African green monkey (Cercopithecus aethiops). It is correctly identified as a green monkey kidney fibroblast cell line
2) Bovine (MDBK): MDBK stands for Madin-Darby Bovine Kidney cells. This cell line is indeed derived from bovine kidney cells

3) Porcine (VR1BL): VR1BL is a fetal porcine retina cell line.

4) Canine (MDCK): MDCK stands for Madin-Darby Canine Kidney cells. This cell line is derived from the kidney of a normal adult Cocker Spaniel dog and is widely used in biomedical research.

5) Bat (EpoNi): The EpoNi cell line is derived from the renal epithelium of bats.

6) Chicken (DF-1): DF-1 is a chicken fibroblast cell line. It is identified as such and is often used in research related to avian biology and virology

Comment 14) Line 447: In the sentence of “...reaching the plateau 3 dpi of roughly 105/ml infectious particles”, 105/ml should be 105 PFU/mL.

Response 14) The missing unit ‘PFU’ was inserted (see line 467)

Comment 15) Fig.4: (a) (b) (c) (d)... are not marked in Figure 4

Response 15) The sub-annotation of the formal Figure 4 (now Figure 5) was indeed missing and was added now.

Reviewer 3 Report

Comments and Suggestions for Authors

Dear colleagues!

After review of the manuscript by Riedl et al. I have the following comments regarding its merit. Generally, it is a high-quality study with sufficient level on technical and scientific rigour. Minor issues are to be addressed by the Authors

1) throughout the text som reference software error has occurred making ref-ce numbers replacement by Error! message

2) please, provide study limitations in Discussion section - the platform has been thoroughly tested in vitro yet until in vivo data is provided one can hardly ensure its benefits.

Best regards, Reviewer.

Author Response

We thank Reviewer 3 for the thorough review of our manuscript. Indeed, we agree that the points you mentioned are of importance and dealing with them improved our manuscript.

Please, find our responses to the raised questions below.

Comment 1) throughout the text some reference software error has occurred making ref-ce numbers replacement by Error! message

We do not know what happened, the field codes for the figures and tables got corrupted. We tried to cure all error-messages and the field code problem too. Hopefully it worked, at least our PDF version looks OK.

Comment 2) please, provide study limitations in Discussion section - the platform has been thoroughly tested in vitro yet until in vivo data is provided one can hardly ensure its benefits.

Indeed, all presented data are in vitro. There were not yet any animal experiments performed, however, certainly it is our ambition to test our vectors in vivo in cross-species settings. However, organizing and carrying out animal experiments, which utilizes models other than rodents are expensive and ethically more demanding and we believe that different models will lead to different conclusions and a number of cross-species application should be tested. Therefore, we believed that it is beneficial for the further development of our MCMV-platform to share our data with the community of vaccine researchers in this early stage. We included a statement pointing of this important limitation of our presented study in the discussion (see lines 607-610).